# CAUSAL IMAGE MODELING FOR EFFICIENT VISUAL UNDERSTANDING

## ABSTRACT

In this work, we present a comprehensive analysis of causal image modeling and introduce the **Adventurer** series models where we treat images as sequences of patch tokens and employ uni-directional language models to learn visual representations. This modeling paradigm allows us to process images in a recurrent formulation with linear complexity relative to the sequence length, which can effectively address the memory and computation explosion issues posed by high-resolution and fine-grained images. In detail, we introduce two simple designs that seamlessly integrate image inputs into the causal inference framework: a global pooling token placed at the beginning of the sequence and a flipping operation between every two layers. Extensive empirical studies demonstrate the significant efficiency and effectiveness of this causal image modeling paradigm. For example, our base-sized Adventurer model attains a competitive test accuracy of 84.0% on the standard ImageNet-1k benchmark with 216 images/s training throughput, which is 5.3× more efficient than vision transformers to achieve the same result.

## 1 INTRODUCTION

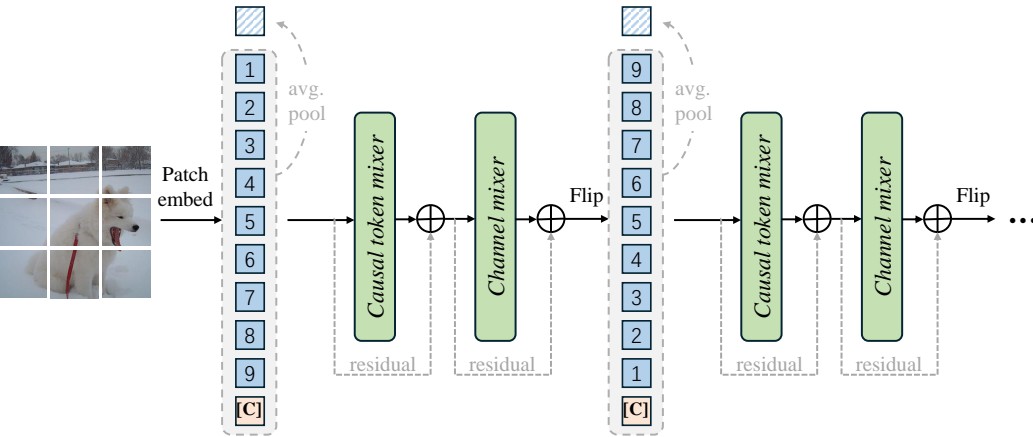

Figure 1: **Causal image modeling framework of Adventurer.** We learn visual representations simply by a uni-directional language model comprising causal token mixers and channel mixers. Two simple designs are introduced to integrate this framework into image inputs: a global pooling token placed at the start of the sequence and a flipping operation between every two causal blocks.

We start introducing our method with a thought experiment: An adventurer holding a torch is exploring an ancient relic located in a dark cave deep in the mountains. A huge mural painted inside the cave has caught his attention. However, the cave is narrow and pitch-dark, with the torch serving as the only source of light, illuminating only a small part of the mural at a time. To figure out what is depicted on the mural, the adventurer has to "scan" it from top left to bottom right. By repeating this process several times, do you think it is sufficient for him to understand the content of the mural?

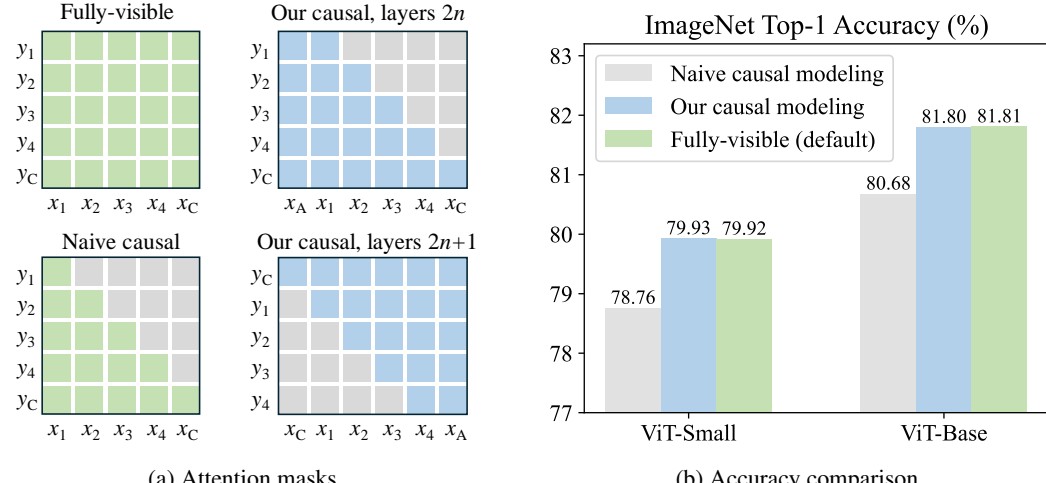

(a) Attention masks                    (b) Accuracy comparison

Figure 2: **Comparison of causal attention formulations.** We compare our causal ViT with the standard ViT where all tokens are fully visible to others and the naive causal ViT with half visibility. (a): Comparison of attention masks. Positions of ▢ denote "invisible". $x_A$ and $x_C$ denote the heading average token and class token respectively. (b): Comparison of accuracy on ImageNet.

In fact, even under good lighting conditions, the area that the human eye can focus on at one time is very limited. For example, when you force your eyesight to focus on **this** word, you cannot count how many words there are in this line. The understanding of complex visual scenes actually relies on the rapid movement of the eyes, known as the *Saccade Mechanism* (Liversedge & Findlay, 2000; Leigh & Zee, 2015). Under this mechanism, the human eye perceives only a very small visual area at a time, and then the eyesight rapidly scans between areas to gain a comprehensive understanding of the whole scene.

This visual understanding mechanism inspires us to consider the possibility of modeling images as 1D sequences of patches. However, in contrast to vision transformers (Dosovitskiy et al., 2021) that require referring all other tokens when understanding each patch token and thereby result in quadratic complexity with respect to sequence length, we aim to employ a more efficient causal modeling approach, which scales linearly with the number of patches and better aligns with the low power consumption characteristic of the human visual system.

To this end, a very straightforward way is dividing the image into non-overlapping patches, flattening them as a 1D token sequence, and then processing them simply by a uni-directional language model. Nonetheless, this naive causal modeling approach is not directly applicable to image understanding because in a uni-directional sequence, each token can only receive information from the tokens before it, which results in tokens at the start of the sequence having poor representations due to the lack of global context.

But interestingly, we find that this issue can be effectively addressed with just two simple designs: first, we place an average pooling token at the beginning of the sequence, which is computed by the average of all other tokens in the sequence. This allows tokens at the start of the sequence to access sufficient global information, thus enhancing the quality of their representations. Second, we introduce sequence flipping operations between layers of the model to counteract the information imbalance caused by positional differences among tokens. We term this two simple designs *heading average* and *inter-layer flipping*, respectively. The overall framework of this Causal Image Modeling (CIM) paradigm is shown in Figure 1.

To validate the efficiency and effectiveness of our CIM framework, we first integrate it with Vision Transformer (ViT) architectures, with Figure 2a showing the difference in attention masks compared to the standard ViT and a naive causal ViT. As we analyzed earlier, causal models can easily impact the representation of tokens at the beginning of the sequence and thereby degrade predictive performance. In Figure 2b, we observe a 1.16% and 1.13% accuracy decrease on ImageNet (Deng et al., 2009) when employing a naive causal ViT-Small and Base respectively. However, this issue is

well addressed by our reformed causal framework, while our causal ViTs can seamlessly match the performance of the standard ones with full token visibility.

This experiment can directly demonstrate the following three points:

1. **Causal modeling is sufficient for image understanding.** We find that when equipped with the heading average token and sequence flipping operations, the uni-directional language models can be directly used for image encoding and achieve results competitive with standard ViTs. We will further elaborate in the following sections that this modeling approach is also applicable to various visual understanding tasks such as semantic segmentation, object detection, and instance segmentation.

2. **The standard ViT involves considerable redundant computations.** As shown in Figure 2, causal modeling ignores around half of the computation in self-attention but can attain almost the same accuracy of standard ViTs. With appropriate parallel processing mechanisms (Dao et al., 2022), causal modeling can substantially accelerate self-attention. For instance, when processing a sequence of 2,000 tokens, causal attention can achieve a speed increase of about 50% compared to the fully-visible attention.

3. **Visual backbones can be much more efficient.** Besides being able to speed up self-attention, the greater advantage of causal modeling lies in the ability to incorporate RNN-like token mixers such as Mamba (Gu & Dao, 2023; Dao & Gu, 2024), whose computation scales linearly with sequence length. Compared to the quadratic complexity of transformers, this unique advantage can effectively solve the problems of computational and memory explosion when processing high-resolution and fine-grained images.

We term our causal image models **Adventurer** to echo the thought experiment at the beginning. As shown in Figure 1, the Adventurer models consist of a number of causal token mixers and channel mixers, incorporating our newly introduced mechanisms of heading average and inter-layer sequence flipping to process image tokens. By default, we use the latest Mamba-2 (Dao & Gu, 2024) structure as the token mixer and SwiGLU MLP (Touvron et al., 2023) blocks as the channel mixer. Empirically, our Adventurer models exhibit strong capabilities in image understanding, showcasing highly competitive results in classification, segmentation, and detection tasks. In dealing with long sequences, our model demonstrates a significant speed advantage. For example, with an input size of $448 \times 448$ and a patch size of $8 \times 8$, *i.e.*, a sequence of over 3,000 tokens, our Base-sized model achieves a competitive test accuracy of 84.6% on ImageNet (Deng et al., 2009) with 5.3 times faster training speed compared with ViT-Base at the same input scale.

## 2 RELATED WORK

**Generic vision backbones.** Convolutional Neural Networks (CNNs) (LeCun et al., 1998) have long been the dominant backbone architecture for various vision tasks. Over time, the CNN structures have experienced a series of major innovations (Krizhevsky et al., 2012; Simonyan & Zisserman, 2015; He et al., 2016; Huang et al., 2017; Tan & Le, 2019; Liu et al., 2022) and now remain competitive to modern visual architectures. Vision transformers (Dosovitskiy et al., 2021), on the other hand, has recently represented a paradigm shift from CNNs' hierarchical feature extraction to patch-by-patch visual encoding. Since the introduction of ViTs, the research community has made substantial strides in developing more robust and efficient training approaches (Touvron et al., 2021; 2022), optimizing model designs (Liu et al., 2021; Chen et al., 2021a; Yuan et al., 2021), and advancing the frontiers of self-supervised learning (Chen et al., 2021b; Caron et al., 2021; Bao et al., 2022; He et al., 2022; Oquab et al., 2023).

**Mamba and state space models.** State Space Models (SSMs) have long been utilized in control systems, primarily to handle continuous inputs (Kalman, 1960). Advancements in discretization methods (Tallec & Ollivier, 2018; Gu et al., 2020; Nguyen et al., 2022; Gu et al., 2023) have expanded the application of SSMs to deep learning, particularly in modeling sequential data (Gu et al., 2022; 2021; Smith et al., 2022). SSMs broadly include any recurrent models that utilize a latent state, ranging from traditional models like Hidden Markov Models (Eddy, 1996) and RNNs to more modern approaches such as Linear Attention (Katharopoulos et al., 2020), RetNet (Sun et al., 2023), and RWKV (Peng et al., 2023). Recently, Gu & Dao (2023) introduce Mamba, a novel SSM block that leverages structured SSMs along with state expansion optimized for hardware efficiency.

**Mamba in visual applications.** Similar to the Transformer's success in NLP and its adoption in vision tasks, Mamba has also been extended to vision fields. For example, Zhu et al. (2024) stack forward and backward scanning blocks to capture bidirectional information, addressing the directionality issue inherent in causal models. Liu et al. (2024b) introduce a hybrid architecture that integrates Mamba with 2D convolution, enabling the capture of contextual information from multiple directions and dimensions. Hatamizadeh & Kautz (2024) utilize a framework that integrates Mamba with self-attention, enhancing the model's capability to capture long-range spatial relationships. Shi et al. (2024) proposes the Visual State Space Duality (VSSD) model, which enhances the performance and efficiency of state space models in sequential modeling tasks by incorporating non-causality and multi-scan strategies. Mamba-based architectures have extended to a wide range of various vision domains (Li et al., 2024; Wang et al., 2024; Yang et al., 2024; Huang et al., 2024; Ren et al., 2024; Lieber et al., 2024; Liu et al., 2024a) .

## 3 METHOD

### 3.1 BUILDING CAUSAL IMAGE MODELS

Overall, we follow the practice of vision transformers (Dosovitskiy et al., 2021) that incorporate patch embedding, positional embedding, token mixers and channel mixers to build our Adventurer models. Formally, given an input image $I \in \mathbb{R}^{3 \times h \times w}$, we first divide it into non-overlapping patches of size $p \times p$, flattening them to form a token sequence $X \in \mathbb{R}^{hw/p^2 \times d}$, where $d$ denotes the number of hidden channels. For easy notations, here we assume $h = w$ and denote $n = h^2/p^2$. Similar to language models, we append the class token at the end of sequence to represent global features. For positional embeddings, here we simply use a learnable matrix $P \in \mathbb{R}^{(n+1) \times d}$ that is added to the patch and class tokens. We leave the exploration of position encoding more suitable for causal models to future works.

Table 1: Configurations of our Adventurer models. Each block consists of a Mamba-2 token mixer and an MLP channel mixer with SwiGLU activation. Here "Dim" denotes the input/output channel dimension of all blocks.

| Model | Blocks | Dim | Params |
|---|---|---|---|
| Adventurer-Tiny | 12 | 256 | 12 M |
| Adventurer-Small | 12 | 512 | 44 M |
| Adventurer-Base | 12 | 768 | 99 M |
| Adventurer-Large | 24 | 1024 | 346 M |

As shown in Figure 1, each layer of our Adventurer model consists of a causal token mixer and a channel mixer. In this work, we discuss two variants of Adventurer models: Transformer-based and Mamba-based. For the former, we use causal self-attention as token mixer and follow the original Vision Transformers (Dosovitskiy et al., 2021) by leveraging a simple Multi-Layer Perceptron (MLP) as channel mixer. For the Mamba-based models, we employ Mamba-2 (Dao & Gu, 2024) as token mixer and MLP with SwiGLU (Shazeer, 2020) activation as channel mixer. Unless otherwise specified, **the term Adventurer by default refers to its Mamba-based variant in this paper.**

We present detailed configurations of our Adventurer models in Table 1. As mentioned above, Mamba (Gu & Dao, 2023) is a RNN-like state space model that has incorporated very efficient hardware-aware designs to support parallel computation. Due to its recurrent formulation, Mamba's computational complexity scales linearly with sequence length, offering an efficient modeling approach that allows us to increase input resolution or reduce patch size to obtain more detailed visual information. For Mamba-2 blocks, we adopt a $2\times$ expand ratio and set the feature dimension to be a multiple of 256 to better leverage its parallel efficiency (Dao & Gu, 2024). Following the recent advances of large language models (Touvron et al., 2023), we opt to use an improved MLP block with SwiGLU activation as channel mixer. We set the hidden dimension of MLP to $2.5\times$ input/output dimension to appropriately reduce the computational load.

### 3.2 ADAPTING IMAGES INTO CAUSAL INFERENCE

As discussed in Section 1, when we use a causal model to process image tokens, it can easily incur the problem of information imbalance. That is, tokens at the end of the sequence can effectively aggregate information from other tokens, while those at the beginning of the sequence struggle

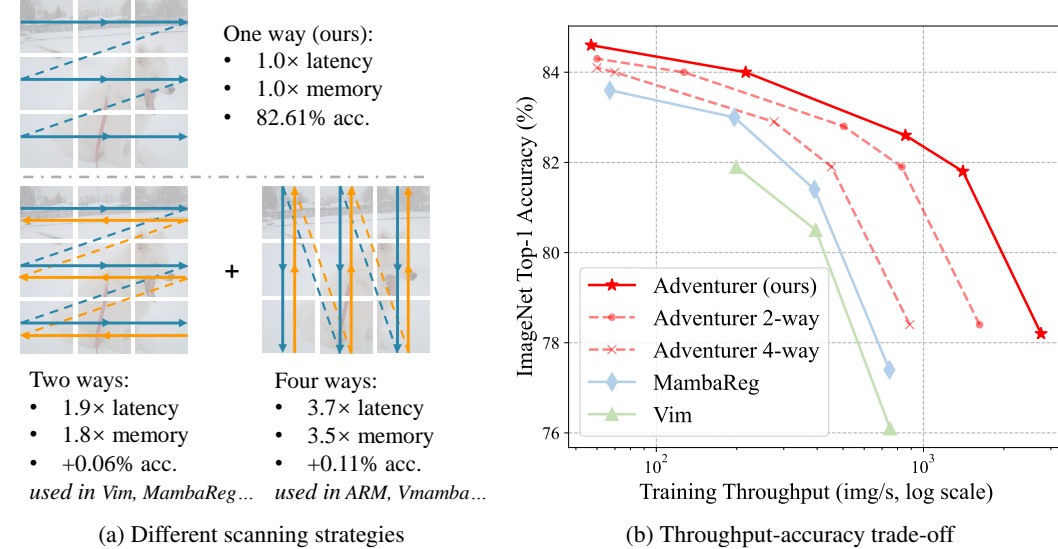

(a) Different scanning strategies    (b) Throughput-accuracy trade-off

Figure 3: **Do we really need multi-way scans for causal image models?** Here we illustrate and compare various scanning strategies, finding that the multi-way approaches hardly offer further performance improvements relative to our models. Please also note that for causal image models involving multi-way scanning, the number of model parameters does not directly reflect its computing cost and actual inference speed as the multiple scans can share most parameters in one layer.

to access the global context of the image. To address this issue, existing causal models typically employ a method of multiple scans and average the multi-way outputs (Zhu et al., 2024; Wang et al., 2024; Liu et al., 2024b). While this method does not significantly increase the number of model parameters, the computation cost and actual inference time are multiplied. Instead, we address the information imbalance problem effectively under the condition of using only one-way scanning in each layer, through two simple mechanisms: Heading Average and Inter-Layer Flipping.

**Heading Average** denotes placing a global average pooling token at the beginning of the input sequence for each Adventurer layer. Formally, given the input for the $i$-th Adventurer block:

$$\boldsymbol{X}^i = \{\boldsymbol{x}_1^i, \boldsymbol{x}_2^i, \ldots, \boldsymbol{x}_n^i, \boldsymbol{x}_{\text{CLS}}^i\}, \;\; \boldsymbol{x}_j^i \in \mathbb{R}^d \;\text{ for }\; j = 1, 2, \ldots, \text{CLS}, \tag{1}$$

we form an augmented sequence by putting an average token at the beginning:

$$\boldsymbol{X}_{\text{aug}}^i = \{\boldsymbol{x}_{\text{AVG}}^i, \boldsymbol{x}_1^i, \boldsymbol{x}_2^i, \ldots, \boldsymbol{x}_n^i, \boldsymbol{x}_{\text{CLS}}^i\}, \;\; \boldsymbol{x}_{\text{AVG}}^i = \frac{1}{n+1} \sum_j \boldsymbol{x}_j^i. \tag{2}$$

This operation forces the average token which contains sufficient global information to be the starting point, enabling the patch tokens at the beginning of the sequence to also access the global context. To ensure that the heading average token accurately represents the global feature for each layer, we discard the output of $\boldsymbol{x}_{\text{AVG}}$ at the end of each Adventurer block and recalculate it by Equation 2 as the next layer's input.

**Inter-Layer Flipping**, instead of inner-layer multi-way scanning, provides with a more efficient strategy to overcome the information imbalance issue. Specifically, our token mixers scan the input sequence only once, and between every two Adventurer blocks, we reverse the order of patch tokens and leave the positions of the class token and average token unchanged. We illustrate different scanning strategies in Figure 3. As is shown, additional scans typically increase the computational load and inference latency proportionally, while our one-way scanning approach possesses optimal efficiency but similar performance compared with multi-way scans. For a clearer understanding of the Adventurer models' computational process, we present a PyTorch-like pseudo code in Algorithm 1.

### 3.3 SPEED AND MEMORY COMPARISON

We quantitatively compare the differences in training speed and GPU memory overhead between Adventurer, ViT, and an existing vision Mamba architecture (Zhu et al., 2024). As shown in Fig-

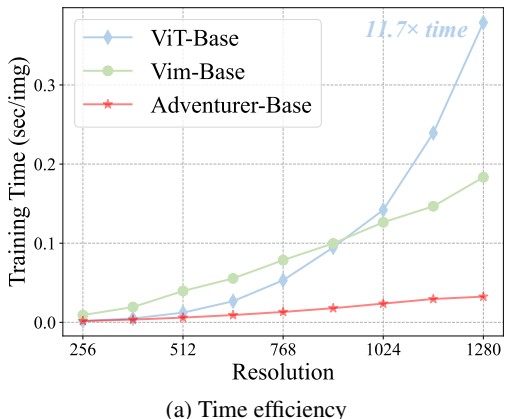 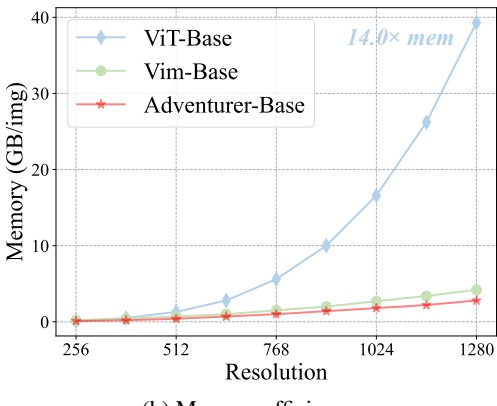

(a) Time efficiency          (b) Memory efficiency

Figure 4: **Time and memory efficiency at different resolutions.** (a): Training time for every one image (second/image) on an A100 GPU. At the input size of $1280^2$, our Adventurer-Base is $11.7\times$ faster than ViT-Base. (b): The memory requirement (GB/image) of Adventurer also grows slowly when input resolution increases, achieving a $14\times$ superior memory efficiency than ViT at $1280^2$.

---

**Algorithm 1** PyTorch-like Pseudocode of Adventurer's Feature Forward Function

---

```
# patch_embed: patchfying input images from [B, 3, W, H] to [B, N, C]
# pos_embed: learnable parameters of positional embeddings, [N+1, C]
# token_mixers: a list of L causal token mixers (by default: Mamba-2)
# channel_mixers: a list of L channel mixers (by default: SwiGLU MLP)

def forward_feature(images):
    x = patch_embed(images) # patch tokens: [B, N, C]
    x = concatenate([x, cls_token], dim=1) # cls at the end: [B, N+1, C]
    x = x + pos_embed # [B, N+1, C]

    for i in range(L): # a total of L blocks
        avg = x.mean(dim=1, keepdim=True) # heading average token
        x = concatenate([avg, x], dim=1) # avg at the start: [B, N+2, C]

        x = x + token_mixers[i](x) # forward token mixer
        x = x + channel_mixers[i](x) # forward channel mixer

        x = x[:, 1:] # discard avg output: [B, N+1, C]
        # flip sequence, leaving cls at the end
        x = concatenate([x[:, :-1].flip(1), x[:, -1:]], dim=1)

    return x
```

---

ure 4, ViT, due to its quadratic complexity relative to sequence length, experiences a rapid increase in processing time and memory requirements as input resolution increases. In contrast, Mamba-based visual backbones, with their linear complexity, exhibit clear advantages in speed and memory efficiency at high resolutions. Remarkably, at an input size of $1280\times1280$, our Adventurer-Base achieves a speed improvement of 11.7 times and a memory savings of 14.0 times compared to ViT-Base. We also note that compared to Vim-Base, our model shows significant speed superiority, a result of our series of structural optimizations: we have implemented efficient one-way scanning, introduced hardware-friendly channel mixers, and adopted the latest Mamba-2 structure.

## 4 EXPERIMENTS

### 4.1 EXPERIMENTAL SETUPS

We primarily evaluate the Adventurer models on the standard ImageNet-1k classification benchmark (Deng et al., 2009). The dataset consists of around 1.28 million training images and 50,000 validation images across 1,000 categories. We follow the recent multi-stage training recipe (Wang et al., 2024) for Mamba models, which comprises 300 epochs pretraining with an input size of $128\times128$, followed by 100 epochs training in $224\times224$ and 20 more finetuning epochs in $224\times224$

| Model | Input size | Params | Throughput (↑) | Memory (↓) | Accuracy (%) |
|---|---|---|---|---|---|
| DeiT-Tiny (Touvron et al., 2021) | 224 | 5M | 3480 | 3.4G | 72.2 |
| DeiT-Small (Touvron et al., 2021) | 224 | 22M | 1924 | 6.8G | 79.8 |
| Adventurer-Tiny (ours) | 224 | 12M | 2757 | 4.2G | 78.2 |
| Adventurer-Small (ours) | 224 | 44M | 1405 | 8.3G | **81.8** |
| DeiT-Base (Touvron et al., 2021) | 224 | 86M | 861 | 14.4G | 81.8 |
| Vim-Tiny (Zhu et al., 2024) | 224 | 7M | 750 | 4.8G | 76.1 |
| MambaReg-T (Wang et al., 2024) | 224 | 9M | 746 | 5.1G | 77.4 |
| ConvNeXt-T (Liu et al., 2022) | 224 | 29M | 635 | 8.3G | 82.1 |
| EfficientNet-B3 (Tan & Le, 2019) | 300 | 12M | 546 | 19.7G | 81.6 |
| Adventurer-Base (ours) | 224 | 99M | 856 | 13.0G | **82.6** |
| ConvNeXt-S (Liu et al., 2022) | 224 | 50M | 412 | 13.1G | 83.1 |
| Vim-Small (Zhu et al., 2024) | 224 | 26M | 395 | 9.4G | 80.5 |
| MambaReg-S (Wang et al., 2024) | 224 | 28M | 391 | 9.9G | 81.4 |
| ConvNeXt-B (Liu et al., 2022) | 224 | 89M | 305 | 17.9G | 83.8 |
| VMamba-B (Liu et al., 2024b) | 224 | 89M | 246 | 37.1G | 83.9 |
| DeiT-Base (Touvron et al., 2021) | 384 | 86M | 201 | 63.8G | 83.1 |
| MambaReg-B (Wang et al., 2024) | 224 | 99M | 196 | 20.3G | 83.0 |
| Adventurer-Large (ours) | 224 | 346M | 301 | 35.5G | 83.4 |
| Adventurer-Base (ours) | 448 | 99M | 216 | 45.2G | **84.0** |
| DeiT-Base/P14 (our impl.) | 448 | 86M | 86 | >80G | 83.5 |
| MambaReg-L (Wang et al., 2024) | 224 | 341M | 67 | 55.5G | 83.6 |
| MambaReg-B (Wang et al., 2024) | 384 | 99M | 63 | 51.4G | 84.3 |
| EfficientNet-B7 (Tan & Le, 2019) | 560 | 66M | 61 | >80G | 84.3 |
| DeiT-Base/P14 (our impl.) | 560 | 87M | 41 | >80G | 84.0 |
| MambaReg-L (Wang et al., 2024) | 384 | 342M | 23 | >80G | 84.5 |
| Adventurer-Base/P8 (ours) | 448 | 100M | 57 | >80G | **84.6** |

Table 2: **ImageNet classification Results**. Instead of comparing theoretical FLOPs, here we report the training throughput and required GPU memory to better reflect the actual time and space complexity of the models. The training throughput (images/second) is tested on an A100 GPU following the protocol of DeiT (Touvron et al., 2021). The memory is tested with a batch size of 128. The terms "Base/P8" and "Base/P14" denotes using smaller patch sizes. We group the baselines by throughput and the best results of each group is **bolded**. Our results are highlighted in blue.

with stronger data augmentation and higher drop path rates. This strategy involves only ∼230 effective training epochs at 224×224 but outperforms the commonly used 300-epoch schedules (Touvron et al., 2021; Zhu et al., 2024). More technical details can be found in the Appendix.

We also evaluate our models in downstream semantic segmentation, object detection, and instance segmentation tasks. For semantic segmentation, we fine-tune our models with an UperNet (Xiao et al., 2018) decoder head on the ADE20k dataset (Zhou et al., 2019), featuring 150 detailed semantic categories within 20K training images, 2K validation images, and 3K test images. Consistent with earlier baselines (Touvron et al., 2021; Zhu et al., 2024), we train the models with a total batch size of 16 across 160,000 iterations, using an AdamW optimizer (Loshchilov & Hutter, 2019) with 5e-5 learning rate and 0.01 weight decay.

We conduct object detection and instance segmentation tasks on the COCO 2017 dataset (Caesar et al., 2018), which consists of 118K training images, 5K validation images, and 20K test images. Following the previous work (Zhu et al., 2024), we employ a Cascade Mask R-CNN (Cai & Vasconcelos, 2019) decoder head and train the model using the AdamW optimizer with a learning rate of 1e-4, a weight decay of 0.05, and a total batch size of 16 for 12 epochs.

## 4.2 MAIN RESULTS

**Image classification.** We evaluate our models on the ImageNet-1k classification benchmark and summarize the results in Table 2. As is shown, our Adventurer models consistently achieve the best predictive performance at each level of training costs, demonstrating the significant efficiency and effectiveness of our one-way causal image modeling paradigm. Remarkably, by increasing the input

| Backbone | Input size | Parameters | Latency (↓) | mIoU (%) |
|---|---|---|---|---|
| ResNet-50 (He et al., 2016) | 512×512 | 64M | 0.69× | 42.0 |
| DeiT-Tiny (Touvron et al., 2021) | 512×512 | 33M | 0.62× | 40.1 |
| DeiT-Small (Touvron et al., 2021) | 512×512 | 51M | 0.68× | 44.0 |
| Adventurer-Tiny (ours) | 512×512 | 20M | 0.50× | 42.1 |
| Adventurer-Small (ours) | 512×512 | 56M | 0.64× | **45.8** |
| ResNet-101 (He et al., 2016) | 512×512 | 83M | 0.74× | 43.8 |
| Vim-Tiny (Zhu et al., 2024) | 512×512 | 16M | 0.77× | 41.2 |
| Vim-Small (Zhu et al., 2024) | 512×512 | 48M | 0.85× | 45.0 |
| MambaReg-S (Wang et al., 2024) | 512×512 | 56M | 0.86× | 45.3 |
| Adventurer-Base (ours) | 512×512 | 115M | 0.86× | **46.6** |
| DeiT-Base (Touvron et al., 2021) | 512×512 | 119M | 1.00× | 45.5 |
| MambaReg-B (Wang et al., 2024) | 512×512 | 132M | 1.42× | 47.7 |
| MambaReg-L (Wang et al., 2024) | 512×512 | 377M | 2.12× | 49.1 |
| Adventurer-Base/P8 (ours) | 640×640 | 115M | 2.04× | **49.4** |

Table 3: **ADE20k semantic segmentation results**. All backbones are pretrained on ImageNet and employ an UperNet decoder head for dense prediction. We group the models by their training latency relative to DeiT-Base. Our results are marked in blue. The best result for each group is **bolded**.

size and decreasing the patch size, we obtain a competitive 84.6% test accuracy with a base-sized model (100M parameters), which showcases the significant impact of scaling up input resolution and reducing receptive granularity on image understanding. This observation also directly underscores the importance of introducing causal image models with linear complexity —— while increasing resolution or reducing granularity substantially aids visual encoding, scaling inputs is particularly challenging for transformers. For example, if we double the width and height of an input image, the length of the token sequence becomes 4× as large. For self-attention, this approximately leads to a 16× computation demand and can easily exceed the limits of existing hardware.

It is equally noteworthy that the Adventurer models are significantly faster than other vision Mamba counterparts such as Vim (Zhu et al., 2024), Mamba-Reg (Wang et al., 2024), and VMamba (Liu et al., 2024b), while this speed improvement primarily originates from our one-way scanning strategy. For example, compared with the previous pure Mamba architecture Vim-Base (98M parameters), our Adventurer-Base (99M parameters) is 3.5× faster yet achieves 0.7% higher test accuracy on ImageNet. More importantly, while most Mamba-based architectures demonstrate a theoretical acceleration over transformers in processing long sequences, their actual speed often falls short when training with the common 196-length short sequences (*i.e.*, the 224×224 input with 16×16 patch size). However, our Adventurer framework makes the first Mamba-based structure whose training speed is comparable to vision transformers in processing such short sequences, and this speed advantage expands as the sequence length increases, achieving about a 15× acceleration over transformers when dealing with sequences around the 3,000-token level.

**Semantic Segmentation.** We further conduct semantic segmentation experiments to validate Adventurer's effectiveness in dense prediction. As summarized in Table 3, our models achieve higher mIoU than the baselines at different levels of training speed. As dense prediction tasks favor higher input resolutions than classification, the speed disadvantage of transformers when processing long sequences becomes evident. Under the default 512×512 input size and 16×16 patch size setup, out Adventurer-Base becomes x.x times faster than DeiT-Base and outperforms it by 1.1% mIoU. Notably, in semantic segmentation task, we are the first to successfully scale up the sequence length to 6400 and achieves a competitive test mIoU of 48.5%, which significantly outperforms Mamba-Reg's 47.7% that consumes similar training cost.

**Object detection and instance segmentation.** We also benchmark the performance of our model on object detection and instance segmentation tasks to further demonstrate its generalizability. Similar to the semantic segmentation task, here our Adventurer models consistently exhibit superior efficiency than CNN, Transformer, as well as Mamba-based models. For example, to attain a bounding-box average precision of 48.4%, the existing SoTA of pure Mamba architecture (Mamba-Reg-Base (Wang et al., 2024)) requires 3.9× training time. By now, the Adventurer models have obtained favorable results in four different visual understanding tasks, showcasing that our causal

| Backbone | Head | $AP^b$ | $AP^b_{50}$ | $AP^b_{75}$ | $AP^m$ | $AP^m_{50}$ | $AP^m_{75}$ |
|---|---|---|---|---|---|---|---|
| ResNet-50 (He et al., 2016) | cascade | 41.2 | 59.4 | 45.0 | 35.9 | 56.6 | 38.4 |
| ReNet-101 (He et al., 2016) | cascade | 42.9 | 61.0 | 46.6 | 37.3 | 58.2 | 40.1 |
| DeiT-Tiny (Touvron et al., 2021) | cascade | 44.4 | 63.0 | 47.8 | 38.1 | 59.9 | 40.5 |
| DeiT-Small* (Touvron et al., 2021) | mask rcnn | 44.7 | 65.8 | 48.3 | 39.9 | 62.5 | 42.8 |
| DeiT-Base* (Touvron et al., 2021) | mask rcnn | 47.0 | 68.2 | 51.4 | 41.8 | 65.1 | 44.9 |
| Vim-Tiny (Zhu et al., 2024) | cascade | 45.7 | 63.9 | 49.6 | 39.2 | 60.9 | 41.7 |
| Adventurer-Tiny (ours) | cascade | 46.5 | 65.2 | 50.4 | 40.3 | 62.2 | 43.5 |
| Adventurer-Small (ours) | cascade | 47.8 | 66.7 | 51.8 | 41.5 | 63.9 | 44.5 |
| Adventurer-Base (ours) | cascade | **48.4** | **67.2** | **52.4** | **42.0** | **64.8** | **45.0** |

Table 4: **COCO object detection and instance segmentation results**. All backbones are pretrained on ImageNet and employ a Cascade Masked R-CNN (Cai & Vasconcelos, 2019) decoder head for the two downstream tasks. Our results are marked in blue. The best results are **bolded**.

| Model | Causal | Heading average | Inter-layer flipping | Accuracy (%) |
|---|---|---|---|---|
| DeiT-Small | ✓ | ✗ | ✗ | 78.8 |
| DeiT-Small | ✓ | ✓ | ✗ | 79.1 (+0.3) |
| DeiT-Small | ✓ | ✗ | ✓ | 79.6 (+0.8) |
| DeiT-Small | ✓ | ✓ | ✓ | 79.9 (+1.1) |
| DeiT-Small | ✗ | ✗ | ✗ | 79.9 (+1.1) |
| Adventurer-Small | ✓ | ✗ | ✗ | 80.3 |
| Adventurer-Small | ✓ | ✓ | ✗ | 80.8 (+0.5) |
| Adventurer-Small | ✓ | ✗ | ✓ | 81.2 (+0.9) |
| Adventurer-Small | ✓ | ✓ | ✓ | 81.8 (+1.5) |

Table 5: **Ablation study of Causal Image Modeling.** We ablate the effect of *Heading Average* and *Inter-Layer Flipping*, which serve as two key components of our causal image modeling framework. We make causal DeiT models by applying the attention mask discussed in Figure 2a. The accuracy denotes ImageNet classification results. Our default setup is highlighted in blue.

image modeling framework facilitates both image-level and pixel-level inference. With significant speed advantages, these experimental results demonstrate the good potential of Adventurer to serve as a foundational visual backbone for future applications, particularly in meeting the practical demands for increasingly high-resolution and fine-grained images.

## 4.3 ABLATION STUDIES

**Components of causal image modeling.** We first ablate the effect of the key components of our causal image modeling framework. As summarized in Table 5, the heading average and inter-layer flipping mechanisms consistently benefit both Transformer and Adventurer models. The table shows that a naive causal modeling is not sufficient to match the baseline performance of existing models, yet our simple designs can effectively address this problem with quite minimal costs. We also observe that the flipping operation typically contributes more to the performance improvement than heading average, which is possibly because flipping not only addresses the information imbalance issue but also facilitates learning direction-invariant features.

**Choices of channel mixers.** As is introduced above, the Adventurer models by default use Mamba as token mixers and SwiGLU MLP as channel mixers. In previous experience, using only Mamba layers (Dao & Gu, 2024; Wang et al., 2024), or employing simple MLPs like those in the original vision transformers (Dosovitskiy et al., 2021; Radford et al., 2021), has also yielded competitive performance on both visual and language tasks. Here we compare these three different designing choices and summarize the results in Table 6. Compared to the Mamba-only architecture, our model shows an overall accuracy improvement of around 0.2%, demonstrating the significant role of introducing a dedicated channel mixers in enhancing model representational capabilities. In addition, since linear layers have better compatibility with hardware, incorporating MLP-like channel mixers also results in a considerable speed increase.

| Mode | Latency ($\downarrow$) | Adv.-Tiny | Adv.-Small | Adv.-Base |
|---|---|---|---|---|
| 24 × Mamba layers | 1.3× | 78.0 | 81.6 | 82.3 |
| 12 × Mamba + 12 × MLP | 1.0× | 78.0 | 81.7 | 82.4 |
| 12 × Mamba + 12 × SwiGLU | 1.0× | **78.2** | **81.8** | **82.6** |

Table 6: **Ablation study of channel mixers.** Here we compare three different choices: a simple MLP with a 4× hidden dimension, a SwiGLU MLP with a 2.5× hidden dimension, and no channel mixer but doubled Mamba layers. The reported results are ImageNet test accuracy (%). "Latency" denotes required training time relative to our default setup which is highlighted in blue.

| #Tokens | Accuracy (%) |
|---|---|
| 1 | **79.5** |
| 4 | **79.5** |
| 9 | 79.3 |

(a) Number of heading tokens

| Token | Acc. (%) |
|---|---|
| duplicate [cls] | 81.4 |
| new token | 81.5 |
| average | **81.8** |

(b) Choice of heading tokens

| Recalculation | Acc. (%) |
|---|---|
| ✗ | 81.4 |
| ✓ | **81.8** |

(c) Recalculation

Table 7: **Ablation study of heading tokens.** We use an Adv.-Small and report ImageNet test accuracy. Results are based on $192^2$ sized inputs for (a), and $224^2$ for (b) and (c). **(a)**: We divide all patch tokens into $N$ parts, averaging them as heading tokens. **(b)**: We compare the effect of duplicating the class token or learning a new one as the heading token. **(c)**: By default, we recalculate the heading average token at the beginning of each token mixer and ablate its impact here.

**Designing the heading token.** To develop Adventurer models, we have tried many different types of heading tokens to facilitate our one-way scanning approach. Here we present these ablation studies in Table 7. First, instead of using a single heading token that is calculated by globally pooling all patches, we can employ relatively finer-grained tokens to represent the global context. In detail, we attempt to equally divide all patch tokens into $N$ grids in the 2D space, averaging the features within each grid to form more and finer heading tokens. However, as summarized in Table 7a, this strategy does not improve performance and may disrupt the original feature distribution, leading to accuracy degradation when there are too many (*e.g.,* 9) heading tokens.

In Table 7b, we also try two more variants that duplicate the class token or train a new learnable token as the heading token, yet both fall short when comparing with our default setup. Additionally, Table 7c proves that it is necessary to recalculate the heading average token at the beginning of each Adventurer layers. These experiments validate our initial hypothesis about the functionality of the heading average mechanism: we need a token that compresses as much global context as possible to serve as the starting point of the sequence. This helps address the issue of tokens at the beginning of the sequence receiving insufficient information. Calculating an average token before the start of each layer proves to be the most straightforward and effective method to accomplish this objective.

## 5 CONCLUSION

In this work, we are motivated to build a simple and efficient visual backbone by causal image modeling. Through extensive empirical studies across various image understanding tasks, we find that by employing two simple mechanisms-——*Heading Average* and *Inter-Layer Flipping*-——causal models can well integrate with visual inputs, matching or even surpassing the predictive performance of traditional vision transformers. Compared to existing vision Mamba architectures (*e.g.*, Vim and MambaReg), our Adventurer framework achieves a significant speed improvement of 4~5 times by simplifying the structure, greatly enhancing the practicality of causal models in vision tasks. Additionally, we demonstrate that our causal image modeling approach is also applicable to transformer architectures, which is able to reduce nearly half of the computational redundancy in self-attention while seamlessly matching the ViT's original performance, showcasing that Adventurer is a generalizable form of image models. In future works, we will fully exploit the speed and linear complexity advantages of Adventurer, experimenting with higher image resolutions or smaller patch sizes to further explore its capabilities in handling very long visual sequences.

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

## A APPENDIX

### A.1 MORE TECHNICAL DETAILS

We basically follow Mamba-Register's three-stage training strategy (Wang et al., 2024) which has been found to be able to effectively prevent Mamba's over-fitting and save training time. The detailed configurations of each stage are shown in Table 8, Table 9, and Table 10, respectively. For the Tiny-sized model, we directly train with the Small/Base's stage-2 recipe for 300 epochs since the training time is short and the tiny model is not easy to get overfit. To train the Large-sized model, we make major modifications of the recipe compared with Mamba-Register to further shorten training time. For all stages, the actually learning rate is calculated by $base\_lr * batchsize/512$; the color jitter factor is set to 0.3; the mixup alpha and cutmix alpha are set to 0.8 and 1.0, respectively.

| Config | Small/Base | Large |
|---|---|---|
| input size | 128 | |
| optimizer | AdamW | |
| base learning rate | 5e-4 | 2e-4 |
| weight decay | 0.05 | 0.3 |
| epochs | 300 | 200 |
| optimizer betas | 0.9, 0.999 | 0.9, 0.95 |
| batch size | 1024 | 4096 |
| warmup epochs | 5 | 20 |
| stochastic depth (drop path) | 0.1 | 0.2 |
| layer-wise lr decay | ✗ | |
| label smoothing | ✗ | |
| random erasing | ✗ | |
| Rand Augmentation | ✗ | |
| repeated augmentation | ✓ | |
| ThreeAugmentation | ✓ | |

Table 8: **Settings of Stage One**

| Config | Small/Base | Large |
|---|---|---|
| input size | 224 | |
| optimizer | AdamW | |
| base learning rate | 5e-4 | 8e-4 |
| weight decay | 0.05 | 0.3 |
| epochs | 100 | 50 |
| optimizer betas | 0.9, 0.999 | 0.9, 0.95 |
| batch size | 1024 | 4096 |
| warmup epochs | 5 | 20 |
| stochastic depth (drop path) | 0.2 (S), 0.4 (B) | 0.4 |
| layer-wise lr decay | ✗ | 0.9 |
| label smoothing | ✗ | |
| random erasing | ✗ | |
| Rand Augmentation | ✗ | |
| repeated augmentation | ✓ | |
| ThreeAugmentation | ✓ | |

Table 9: **Settings of Stage Two**

| Config | Small/Base | Large |
|---|---|---|
| input size | 224 | |
| optimizer | AdamW | |
| base learning rate | 1e-5 | 2e-5 |
| weight decay | 0.1 | 0.1 |
| epochs | 20 | 50 |
| optimizer betas | 0.9, 0.999 | 0.9, 0.95 |
| batch size | 512 | 512 |
| warmup epochs | 5 | 5 |
| stochastic depth (drop path) | 0.4 (S), 0.6 (B) | 0.6 |
| layer-wise lr decay | ✗ | 0.95 |
| label smoothing | 0.1 | |
| random erasing | ✗ | |
| Rand Augmentation | rand-m9-mstd0.5-inc1 | |
| repeated augmentation | ✗ | |
| ThreeAugmentation | ✗ | |

Table 10: **Settings of Stage Three**

