# OpenReview forum: "Causal Image Modeling for Efficient Visual Understanding"
_ICLR.cc/2025/Conference — ICLR 2025 Conference Withdrawn Submission_

### Official Review · Reviewer_Xuoy · 2024-11-02

**Soundness:** 3
**Presentation:** 3
**Contribution:** 2
**Rating:** 5
**Confidence:** 4

**Summary:**

The paper proposes a variation of causal attention (state-space-models?) for vision transformer named Adverturer. To this end, they propose an architecture that flips the direction of the attention operation for each layer and further adds a global pooling token at the beginning of each sequence. The approach is evaluated on ImageNet classification and ADE20k semantic segmentation tasks.

**Strengths:**

- The topic of efficient attention is of high interest to the community.

- The paper is well-written and easy to follow.

- The figures are clear and comprehensive.

**Weaknesses:**

- 1. Significant originality of ideas and/or execution - Part1

To my understanding, the proposed architecture is mainly a variant of a bidirectional Vision Mamba, just that the bidirectional attention is done in two separate layers, with token and channel mixers in between (if I missed something here, please correct me). The impact of this variation seems to be very marginal (see Tab. 6). So I'm not sure if this could be considered a significant novelty.

- 2. Significant originality of ideas and/or execution - Part2

The second contribution, which I would consider novel (at least didn't see it before), is the average pooling token, which seems to give a slight increase (+0.5 on INET), but this might be too incremental for a top A conference.

- 3. Performance on SotA

I understand that the field of Vision Mamba variations is moving fast and newer models are coming up every month (https://arxiv.org/abs/2404.18861), but even by just applying the SotA table provided by the authors, I don't see a significant benefit of the architecture. Eg. in Tab. 2:  Adverturer-base (99M Params) = 82.3% compared to VMamba-B (89M params) = 83.9. To get to 84%, Adverturer-base (99M Params) actually needs 448 instead of 224 input resolution and thus more memory. I could list more examples, but overall, I don't see a clear advantage here over regular Vision Mamba architectures.

**Questions:**

- Why don't you call it Mamba or state place model if that's what it is (again, if I missed something here, please correct me)?

- Could you comment on the novelty and performance issue?

---

### Official Review · Reviewer_LqDY · 2024-11-04

**Soundness:** 2
**Presentation:** 3
**Contribution:** 2
**Rating:** 5
**Confidence:** 3

**Summary:**

This paper introduces Causal Image Modeling with the "Adventurer" model, which applies causal modeling to image patch sequences. The key components are a global pooling token and layer-wise flipping (to improve context representation). Experiments show that Adventurer performs comparably or better than Vision Transformers on benchmarks such as ImageNet and COCO.

**Strengths:**

+ This paper proposes an efficient causal approach to image modeling. It achieves competitive ImageNet accuracy while reducing computational complexity, which is beneficial for high-resolution and fine-grained images.

+ The Adventurer models perform well across tasks like classification, segmentation, and detection, suggesting suitability as a versatile vision model. However, the efficiency gains rely on specific design mechanisms, limiting adaptability across broader architectures and real-world applications.

**Weaknesses:**

1. Missing important comparisons. The paper introduces pooling tokens and token flipping in vision transformers to enhance efficiency but lacks comprehensive comparisons with related methods. For example, PoolFormer[1] employs pooling as the token mixer, achieving 82.1% accuracy on ImageNet with 11.6 GMACs. At the same time, Shuffle Transformer[2] applies a more generalized token shuffling operation to improve sequence mixing, reaching 84.0% accuracy with a throughput of 279 images per second on a V100 GPU. Including direct comparisons with these methods in Table 2 would clarify Adventurer’s unique contributions and the specific performance advantages brought by these new operations.

2. Limited evaluation. Table 5 provides an ablation study on the effects of the heading average and inter-layer flipping mechanisms, but it is limited to small-parameter versions of Transformer-based and Mamba-based models and only evaluates accuracy. Expanding Table 2 to include results for both Transformer-based and Mamba-based models across multiple parameter scales would provide a more comprehensive assessment of each architecture's strengths and resource efficiency.

3. Insufficient justification. The paper could benefit from clearer explanations of how mechanisms like pooling tokens and inter-layer flipping lead to performance improvements. Strengthening the connection between these design choices and their impact would enhance clarity and reader understanding.

**Questions:**

While the paper presents its approach as "Causal Image Modeling", the method essentially decomposes fully-visible bidirectional attention into partially-visible unidirectional layers, which is an overall bidirectional structure. A significant advantage of causal attention is its ability to cache past computations, allowing new tokens to be processed without re-computing earlier token features, thereby improving deployment efficiency in iterative scenarios. However, the use of pooling tokens and flipping operations requires reprocessing all tokens across layers when new tokens are introduced, potentially limiting the caching benefits and negatively affecting the deployment efficiency in real-time applications. Would the author justify these?

---

### Official Review · Reviewer_a9Y6 · 2024-11-04

**Soundness:** 2
**Presentation:** 2
**Contribution:** 2
**Rating:** 5
**Confidence:** 3

**Summary:**

This paper proposes a novel causal image modeling paradigm called Adventurer. It is based on the latest Mamba-2 architecture and introduces two new techniques: inserting a global average token at the head of the sequence (head-average) and flipping the input sequence between different layers of the model (inter-layer flipping). Experimental results demonstrate that causal models can achieve comparable performance to non-causal ViTs while saving more redundant computations, thereby being more efficient.

**Strengths:**

1. The experimental results are robust, demonstrating the efficacy of the Adventurer models across various tasks such as classification, segmentation, and detection.

2. The ablation studies are comprehensive, showing the results of adopting different design choices, including the causal image modeling components (heading average, inter-layer flipping), channel mixers and heading token design.

**Weaknesses:**

1. The motivation and background are not well discussed in the introduction. For example, Why 1-D causal modeling is necessary for visual images? Although humans can only focus on limited regions at one time, we can freely switch focus across different spatial positions, which causal modeling with fixed image patches cannot achieve. This makes the motivation somewhat inadequate.

    Additionally, a major advantage of 1-D causal modeling is its autoregressive token-by-token generation, supporting decode-only structures like GPT. However, the token flipping introduced in the paper compromises this key feature, raising further questions about the motivation of using 1D modeling in this context.

**Questions:**

Given that images can also support causal modeling now, I wonder what its performance will be on image-text tasks such as image caption and image retrieval.

---

### Official Review · Reviewer_LBWf · 2024-11-04

**Soundness:** 2
**Presentation:** 3
**Contribution:** 2
**Rating:** 3
**Confidence:** 4

**Summary:**

The paper introduces the Adventurer series models for causal image modeling, which treats images as sequences of patch tokens processed by uni-directional language models. Key innovations include a global pooling token at the beginning of the sequence and a flipping operation between layers. Empirical results show that the base-sized Adventurer model achieves a test accuracy of 84.0% on the ImageNet-1k benchmark, with a training throughput of 216 images per second.

**Strengths:**

- This paper is clearly written and easy to follow.
- The intuition of design is interesting and convincing.
- The experiments are thorough.

**Weaknesses:**

- It would be helpful to add the FLOPs metric when comparing with state-of-the-art methods, as most related works typically report this information.
- Please clarify the differences between the proposed method and VMamba. A detailed comparison would help highlight the unique contributions of your approach.
- In Table 2, VMamba achieved an accuracy of 83.9% with fewer parameters and less memory, while also providing a higher output, and the input image size is smaller than that of the Adventurer-Base model. Could you explain these results and the implications for your model?
- The analysis should incorporate more state-of-the-art methods, such as LocalMamba [1], to provide a broader context for your findings and strengthen the comparison.


[1] Huang T, Pei X, You S, et al. Localmamba: Visual state space model with windowed selective scan[J]. arXiv preprint arXiv:2403.09338, 2024.

**Questions:**

See above.

---

### Note · Authors · 2024-11-12

I have read and agree with the venue's withdrawal policy on behalf of myself and my co-authors.